# School-based physical activity interventions among children and adolescents in the Middle East and Arabic speaking countries: A systematic review

Abdullah Alalawi[1]*, Lindsay Blank[2], Elizabeth Goyder[2]

1 Health and Related Research, University of Sheffield, Sheffield, United Kingdom, 2 School of Health and Related Research, University of Sheffield, Sheffield, United Kingdom

☯ These authors contributed equally to this work.
* Aoalalawi1@sheffield.ac.uk

**Data Availability Statement:** All relevant data are within the paper and its Supporting information files.

## Abstract

### Background

It is widely recognised that noncommunicable diseases are on the rise worldwide, partly due to insufficient levels of physical activity (PA). It is a particularly concerning health issue among children and adolescents in Arabic countries where cultural and environmental factors may limit their opportunity for engaging in physical activities.

### Aim

This review sought to assess the effectiveness of school-based PA interventions for increasing PA among schoolchildren aged six to 18 years in Middle Eastern and Arabic-speaking countries.

### Methods

A systematic literature search was developed to identify studies reporting the evaluation of school-based PA interventions in Arabic-speaking countries. Four different databases were searched from January 2000 to January 2023: PubMed/MEDLINE, Web of Science, Scopus and CINAHL. Article titles and abstracts were screened for relevance. Full article scrutiny of retrieved shortlisted articles was undertaken. After citation searches and reference checking of included papers, full data extraction, quality assessment and narrative synthesis was undertaken for all articles that met the inclusion criteria. This review adhered to PRISMA guidelines for conducting systematic reviews.

### Results

Seventeen articles met the inclusion criteria. Eleven articles reported statistically significant improvements in the levels of PA among their participants. Based largely on self-reported outcomes, increases in PA between 58% and 72% were reported. The studies with a follow-up period greater than three months reported sustained PA levels. There are a limited range

**Funding:** This review is a part from PhD study and the first author was sponsored by the Saudi Arabia government but didn't get a specific fund for this review. Also, the publication fees will be covered by the University of Sheffield.

**Competing interests:** The authors have declared that no competing interests exist.

of types of programmes evaluated and evaluations were only identified from 30% of the countries in the region. Relatively few studies focused solely on PA interventions and most of the interventions were multi-component (lifestyle, diet, education).

## Conclusions

This review adds to the existing body of research about the efficacy of school-based interventions to increase physical activity levels. To date, few evaluations assess PA specific interventions and most of the interventions were multi-component including education components on lifestyle and diet. Long-term school-based interventions combined with rigorous theoretical and methodological frameworks are necessary to develop, implement and evaluate PA interventions for children and adolescents in Arabic-speaking countries. Also, future work in this area must also consider the complex systems and agents by which physical activity is influenced.

## Introduction

It is widely recognised that noncommunicable diseases (NCDs) are on the rise worldwide, partly due to insufficient levels of physical activity (PA) [1]. PA is considered one of the best methods for preventing cardiovascular diseases and it can also improve muscular strength and balance, as well as heart and lung function [2]. Taking part in regular PA reduces the likelihood of many diseases such as obesity, diabetes, and high blood pressure [2], and it is critical [3–5] for managing non-communicable conditions and diseases such as hypertension, heart diseases and some cancers. It improves and maintains healthy body weight, mental health, quality of life and general well-being [6]. In developed countries, current levels of PA participation are generally lower than recommendations for gaining health benefits for both adults and children [7, 8]. With specific regard to children and adolescents, insufficient PA during any stage of growth can greatly contribute to being overweight and obese, since these life stages are crucial periods in the acquisition of healthy habits [7]. For this period, physical activity helps to build stronger muscles and bones, improve cardiovascular fitness, control weight and reduce symptoms of depression and anxiety [9].

Insufficient PA may increase the likelihood of higher cholesterol and blood pressure and lead to the development of chronic diseases during adulthood [7]. Children and adolescents who are not doing enough physical activity are more vulnerable to energy imbalance (i.e. consuming more energy through their diet than they expend through PA), thereby increasing their risk of becoming obese or overweight and developing a number of associated complications. More specifically, Chaabane et al., [10] reported that Arabic countries has the highest concentration of non-communicable diseases in the world and have the second-highest prevalence of diabetes worldwide (10.9%) [9] and they are recording a speedy increase in the level of obesity [10, 11]. This low level of PA is one of the factors that has contributed to a pandemic of chronic diseases in the region, such as heart disease, obesity and diabetes [12]. Arabic speaking countries located in the Arabian Peninsula, northern parts of Africa and East Mediterranean [13]. It contains 22 countries ((i.e. Yemen, United Arab Emirates, Syria, Saudi Arabia, Qatar, Oman, Lebanon, Kuwait, Jordan, Bahrain, Palestinian, Tunisia, Egypt, Libya, Morocco, Comoros, Djibouti, Mauritania, Algeria, Sudan, Somalia and Iraq). These countries defined as the 22 member countries of the League of Arab States [14].

It has been acknowledged that physical inactivity is the fourth leading risk factor for world-wide mortality, and it can cause nearly three million deaths globally [15]. It is one of the fourth main contributors to health risk in Arabic countries [16]. Studies [17, 18] report a disproportionately high prevalence of high body mass index (BMI), high blood pressure and cardiovascular disease in the Middle Eastern and Arabic-speaking countries. Other more specific studies [19, 20] note that female children and adolescents are more largely represented in this population (high BMI, blood pressure, and incidence of cardiovascular disease) than their male counterparts. There is also a consensus among researchers that these non-communicable diseases can be attributed to the sedentary behaviour that has resulted from economic development and social changes, and to a conservative culture that determines gender roles. Physical inactivity is most prevalent in the Middle East region (35%) and Saudi Arabia has the highest rate of physical inactivity among members of the Gulf Cooperation Council. In particular, 58.8% of Saudi adults are considered physically inactive [15]. A major consequence of physical inactivity is that the complications associated with the diseases that develop from sedentary lifestyles can carry on into adulthood and significantly compromise individuals' quality of life.

Children and young people aged between six and 18 years represent a considerable proportion of the total population of Middle Eastern and Arabic-speaking countries [18]. Considering that this group is easily accessible in large groups and for considerable periods during the school day, it is a common setting for testing the effectiveness, feasibility and acceptability of interventions aimed at increasing PA.

We therefore undertook a systematic review to assess the effectiveness, feasibility and acceptability of school-based interventions in increasing PA among schoolchildren aged six to 18 years in Middle Eastern and Arabic-speaking countries. We also aimed to identify the key components required in a school-based intervention programme to increase PA among this target population and to understand if there are particular combinations or components of school-based programmes that are more effective than others in increasing physical activity.

## Methods

A systematic literature search and narrative synthesis was undertaken to identify, evaluate and synthesis relevant evidence from intervention studies undertaken in a school setting in Arabic speaking countries. This review adhered to PRISMA guidelines for conducting systematic reviews. The protocol was registered in the PROSPERO international prospective register of systematic reviews (registration number: CRD42021260738).

### Inclusion and exclusion criteria

The inclusion criteria were based on the PICO framework: population (P): school children and young people aged 6–18 years old in Middle Eastern and Arabic-speaking countries, intervention (I) school-based PA as either a primary intervention or a component of a multi behavioural intervention, comparison (C) no intervention or usual school provision of PA and outcome (O) self-reported or objectively measured change in PA levels.

We included any study that met the following criteria: (1) studies that focused on physical activity/inactivity among children and young people in Middle Eastern and Arabic-speaking countries located in the Arabian Peninsula, northern parts of Africa, and the East Mediterranean [13]. It contains 22 countries ((i.e., Yemen, United Arab Emirates, Syria, Saudi Arabia, Qatar, Oman, Lebanon, Kuwait, Jordan, Bahrain, Palestinian, Tunisia, Egypt, Libya, Morocco, Comoros, Djibouti, Mauritania, Algeria, Sudan, Somalia, and Iraq). These countries are defined as the 22 member countries of the League of Arab States [14]; along with countries in the Middle East that are not counted in the Arabic-speaking countries (i.e., Iran, Israel, Turkey,

and Cyprus). (2) children and young aged six to 18 years. (3) studies that explored at least one intervention for increasing physical activity among the targeted population. (4) studies published after the year 2000. (5) studies published in English. (6) school-based interventions.

We excluded any interventions not based in schools and any study targeting specific populations based on BMI or other risk factors for obesity. Any study conducted outside of the Arabic-speaking countries were excluded.

**Search strategy.** Systematic searches were conducted in PubMed/MEDLINE, Web of Science, Scopus and CINAHL and these databases were chosen not only because they are easily accessible but also because they contain a large volume of empirical and peer-reviewed articles, due to their public health scope. The search strategy (S1 Appendix in S1 File).

The search parameters were limited to English and Arabic language publications published between January 2000 and January 2023.

## Selection of studies

All records are exported to Mendeley reference manager software. Duplicate records and all studies whose title showed did not meet the inclusion were excluded. Screening of studies was conducted by two reviewers independently, titles and abstracts of records retrieved from searches were screened for inclusion by two of them, and any disagreement was resolved by discussion with a third reviewer.

## Data extraction

A standardised approach was used to extract data from the selected studies, [21], including study details (author, year of publication and the country where the study conducted), sample description (i.e. sample size, age, sex), interventions (i.e. PA component, combination of PA components), intervention duration, frequency of the intervention, outcomes measured, implementers and study results.

## Risk of bias analysis

Two critical appraisal tools were used to assess the risk of bias within the studies included in this review (the CASP RCT Checklist and the Joanna Briggs Institute (JBI) Quasi-Experimental Studies Tool [22, 23]). The assessment aimed to clearly understand the studies' strengths, weaknesses, and limitations, particularly regarding study design and choice of outcome measures.

## Data synthesis

Due to the heterogeneous nature of the interventions and outcome measures, a meta-analysis approach to the retrieved studies was considered inappropriate. The effectiveness of the PA intervention was assessed by comparing the differences between the intervention groups and control groups immediately after the intervention and during the follow-up periods. The following subcategories were analysed: (1) effectiveness of the PA intervention, (2) key components in the PA intervention and (3) combinations of school-based programmes and their effectiveness on PA.

# Results

## Search results

The search was conducted in January 2023. The initial literature search produced a total of 1526 studies. Further articles (n = 9) were identified from reference lists. The total number

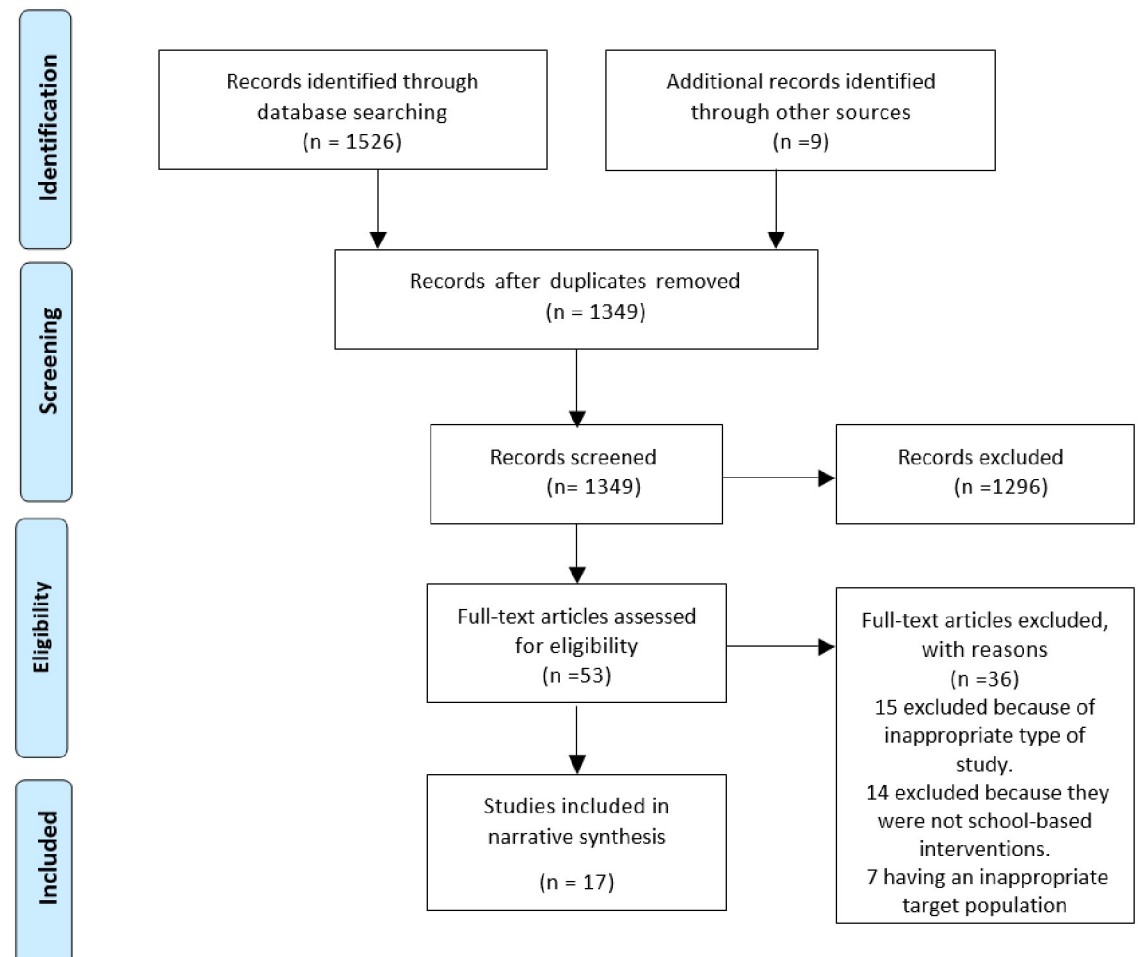

**Fig 1. PRISMA flow chart for study selection.**

was 1535: PubMed/MEDLINE 399, Web of Science 767, Scopus 7 and CINAHL 353. There were 186 duplicates, and after their exclusion, the 1349 remaining records' titles and abstracts were screened against the inclusion and exclusion criteria of the systematic review. The full texts of the remaining 53 studies were screened which led to the exclusion of 36 studies due to such reasons as (1) having an inappropriate target population; (2) being ongoing studies; (3) having inappropriate designs; and (4) inappropriate interventions (see S2 Appendix in S1 File for excluded studies). A PRISMA flow diagram shows search results (Fig 1).

## Study characteristics

Study characteristics as summarised in Table 1. The designs of the 17 studies were quasi-experimental, i.e., non-randomised control trials (four studies) [24–27] or randomised controlled trials (RCTs) (13 studies) [28–40]. Studies were conducted in only seven countries, with no studies from the majority of the Middle East and North Africa (MENA) countries. Three studies were carried out in Tunisia [24–26], five studies in Iran [28, 30, 32–34] three studies in

**Table 1. Summary of the included studies.**

| Study and Year | Country | Study design | Sample size (N) | Gender & Age | Type of Intervention | Length of Intervention & Follow up | Intervention Components | Outcomes Measured | Study Results |
|---|---|---|---|---|---|---|---|---|---|
| Maatoug et al., 2015 [25] | Tunisia | Quasi-Experimental Intervention Study (Non-randomised Control Trial) | 4003 IG 1929 CG 2074 | Both (11 to 16 y) | PA and healthy eating programme. | Three years No follow-up | Educational lessons on PA + how it can incorporate into daily exercises (one hour for three days per week) Peer education through training of student leaders. | PA standardised, pretested questionnaire. | Increase in normal weight (P = .03) Decrease in overweight (P = .03) |
| Maatoug et al., 2013 [24] | Tunisia | Quasi-Experimental Intervention Study (Non-randomised Control Trial) | 2200 IG 1247 CG1091 | Both (12 to 16 y) | PA and healthy eating programme. | One school year. Follow-up after one year. | Intervention contained in interactive lessons and activities that were delivered by pre-formed teachers with the collaboration of the doctor members of the project. | Behaviours Knowledge PA and dietary habits intentions. | Significant increase in both knowledge and behaviour intentions in intervention group (P < .01). |
| Habib-Mourad et al., 2020a [37] | Lebanon | RCT | 1013 IG 701 CG 312 | Both (9 to 11 y) | Focused on dietary behaviours and PA. | One year No follow up | Three modules coordinated the intervention, 12 fun and interactive classroom sessions, family module includes meetings, health fairs, and recipes, and samples sent home as part of the information packets and intervention involved eating at school shops and using lunch boxes sent by families | Knowledge Self-efficacy PA and dietary behaviours | Significant increase in knowledge and self-efficacy in intervention group but PA didn't improve. |
| Habib-Mourad et al., 2020 [36] | Lebanon | RCT | 806 IG 457 CG 349 | Both (8 to 12 y) | Focused on the promotion of healthy eating and an active lifestyle based on social cognitive theory. | Two-year intervention program followed by a one-year washout period. | Two consecutive academic years: Twelve nutrition education interactive activities were delivered in the classroom during the first academic year, and six complimentary activities were delivered during the next 8 months. | Pre-and post-assessment anthropometric measurements and questionnaires about PA and eating habits. | Significance increases in PA after school in the intervention group compared to control group after one year follow-up (P = 0.044) |
| Habib-Mourad et al., 2014 [31] | Lebanon | RCT | 363 IG 188 CG 175 | Both (9 to 11 y) | Multi-component school-based intervention programme focuses on PA and eating. | Three months No follow up | The intervention is called "Kanz al Soha" in Arabic language, which means the treasure of health—focusing on PA and diet. | WC BMI Knowledge PA and dietary habits | There is no significance difference in PA and screen time nor in BMI. Highly acceptance programme. Significant increase in self-efficacy and knowledge. |

(*Continued*)

**Table 1.** (Continued)

| Study and Year | Country | Study design | Sample size (N) | Gender & Age | Type of Intervention | Length of Intervention & Follow up | Intervention Components | Outcomes Measured | Study Results |
|---|---|---|---|---|---|---|---|---|---|
| El Ansari et al., 2010 [29] | Egypt | RCT | 160 IG 80 CG 80 35 boys 45 girls | Both Mean age 15.7 | PA intervention programme. | Three months and conducted after-school hours. No follow up | One hour, three times a week. Outdoor activities such as football, volleyball, handball, basketball, cycling and walking. Indoor activities: table tennis, gym, aerobic dance, and aerobic boxing. | Primary outcome: change in PA. Secondary outcome: change in the anthropometric (BMI, weight, % of body fat). | The weight increased by 37.3% in the control group and decreased by 12.5% after three months in the intervention group. |
| Stanley et al., 2017 [35] | United Arab Emirates | RCT | 394 | Both mean age 13.9 y. | Workshops for health education intervention on tobacco, nutrition and PA. | Not stated | Worksop focuses on health education for PA, nutrition and tobacco. | Attitudes, knowledge, and perceptions. | No change at all in measured parameters. Only change after tobacco workshops. |
| Ghammam et al., 2017 [26] | Tunisia | Quasi-Experimental Intervention Study (Non-randomised Control Trial) | 4003 | Both (11 to 16 y) | PA educational program. | Three-year intervention One year follow up | Three events each year to teach children, parents, and teachers about healthy eating and PA. Interactive lessons about healthy eating, the benefits of regular PA, and how to incorporate PA into daily life. | PA habits and smoking habits. BMI Vegetables and Fruits consumption. | Significance decreases in the students who did PA in the intervention group. (Maybe because the matter of the Tunisian Revolution that happened in the middle of the project) (P = .01, P = .001) |
| Rostami-Moez et al., 2017 [33] | Iran | RCT | 344 IG 179 CG 165 | Girls 7th-grade | Educational sessions for PA and health eating. | Intervention for two months and follow-up for six months. | 60 minutes of training per week for two months (8 hours total) in the field of PA pyramids. Several methods were employed, including lecture, group discussion, poster (PA pyramid), and pamphlets | Self- reported PA | Significance increases in PA in the intervention group from 2.50 to 3.17 |
| Taymoori & Lubans (2008) [28] | Iran | RCT | 166 THP 59 HPM 54 CG 53 | Girls Grades 9 and 10 | Educational sessions for PA and two PA sessions. | Six months Follow-up phone calls to support behaviour change. | Each session 45 to 60 minutes Lectures, role-playing, slides, reminder cards, physical activity planning, and handouts. | Self- reported PA | Significance increases in PA through changes in the theoretical constructs. |
| Darabi et al., 2017 [32] | Iran | RCT | 578 IG 289 CG 289 | Girls (12 to 16 y) | Educational sessions for PA and health eating. | 6 months 6 months follow up | 90 minutes for group discussions. Sessions contain lectures, role play, slides, and educational pamphlets. | Questionnaire to evaluate PA related attitudes, knowledge, Subjective norm Behaviour intention | A significant increase in the intervention group compared to control group (p<0.001) |

*(Continued)*

**Table 1.** (Continued)

| Study and Year | Country | Study design | Sample size (N) | Gender & Age | Type of Intervention | Length of Intervention & Follow up | Intervention Components | Outcomes Measured | Study Results |
|---|---|---|---|---|---|---|---|---|---|
| Saffari et al., 2013 [30] | Iran | RCT | 365 IG 186 CG 179 | Both (14 to 18 y) | Educational sessions for PA and health eating. | Three months and three months follow-up. | Healthy lifestyle course consisting of six sessions over three weeks. | The adolescent Lifestyle Questionnaire | The boys had greater levels of PA than girls. Girls have more social support then boys. |
| Simbar et al., 2017 [34] | Iran | RCT | 80 IG 80 CG 80 | Girls (14 to 18 y) | Educational sessions for PA and health eating. | Not specified how long the intervention. Two months follow-up. | 15-minute role-play, followed by a 15-minute small group discussion on the benefits of RPA 15-minute game-playing exercise then followed by a 15-minute problem-solving exercise | Questionnaire to evaluate PA related attitudes, knowledge and behaviour. | A significant increase in knowledge and behaviour but not in the attitude in the intervention group. |
| Kutbi et al., 2019 [38] | KSA | Cluster RCT | 148 IG 79 CG 69 | Male students (10 to 15 y) | Educational program | 2 months 2 months follow-up | 60 minutes session in health education for 2 weeks. 2 weeks group counselling. Presentation about healthy diet. | PA frequencies (e.g., running, walking, cycling, days/week, how much time spent in sedentary behaviours. | There is no statistically significant differences in total METs between intervention group (2098.41±1922.67) and control group (2216.46 ± 1816.03), P>0.05. |
| Allafi 2020 [39] | Kuwait | RCT | 225 110 boys 15 girls | Both (9–11 y) | PA | 6 weeks 6 weeks follow-up | Reward for walking, advice on pedometers and incentive of ten stickers if 3000 steps achieved. 5x50 minute exercise sessions | Steps/day | The feedback and reward group showed significantly higher steps: 3,429 (SD 458), than the feedback group: 2,655 (SD 577) and control group: 2,091 (SD 483). Also, there is no difference between girls and boys. |
| Elfaki et al., 2020 [27] | KSA | Quasi-Experimental Intervention Study (Non-randomised Control Trial) | 565 | Girls (12–15 y) | Educational sessions for PA and health eating. | 6 months No follow up | Heath education sessions for PA and eating, lectures, discussion, games and role playing. | Self- reported PA Anthropometric measurement. | Significant improvement in PA. |
| Bahathig & Saad (2022) [40] | KSA | Cluster RCT | 138 IG = 68 CG = 70 | Girls (13–14 y) | Educational intervention | 3 months 3 months follow up | 60 minutes seminar for the mothers of girls before the education. Six educational sessions on nutrition, PA and BI through the year. | Self- reported PA, Sedentary behaviour and body Image Satisfaction | There was no significant change in BMI but there are immediate significantly improve in PA and SB. |

Lebanon [31, 36, 37], three studies in Saudi Arabi [27, 38, 40], one study in Egypt [29], one study in Kuwait [39] and one study in the United Arab Emirates (UAE) [35].

Only five studies had fewer than 200 participants [34] (n = 80), [40] (n = 138), [38] (n = 148) [29] (n = 160), [28](n = 166).

Four studies had more than 1,000 participants [24–26, 37]. All the studies focused on school aged children as their target population with a range from 8 years [36] to 16 years [24, 25].

Calculating the overall mean age of the 15,413 participants accounted for in the 17 studies was difficult because the individual data was not available. A majority of the 17 studies included both male and female adolescent students (n = 10), six studies only included girls [27, 28, 32–34, 40], and one study only included boys [38].

All 17 studies specified their intervention periods except two studies [34, 35]; they ranged from six weeks [39] to three years [25, 26] Seven studies used usual practice (or no PA programme) as their control intervention [24–27, 36, 39, 40]. Five studies did not specify their comparison elements, but it can be assume it was a comparison with no physical activity modification program [30, 31, 33, 34, 36]. Four studies used physical activity and physical activity education only as the control since the intervention groups integrated physical activities with other measures including nutrition classes for children, parents and teachers among other measures [28, 29, 32, 38]. One study used a tobacco workshop as its control [35].

## Risk of bias assessment

Risk of bias assessment found that eleven out of the thirteen studies [28, 29, 31–40] classified as RCTs had low risk of bias and one study [30] had an unclear risk of bias. Risk of selection bias was essentially low as only one study [30] did not specify their random sequence generation approach. All other thirteen studies classified as RCTs applied adequate random sequence generation. The risk of reporting bias, on the other hand, was high as for the thirteen trials blinding was not mentioned in their studies and we can assume blinding was not possible. However, incomplete data were sufficiently accounted for in all thirteen trials. Furthermore, selective bias was low in eleven trials and unclear in only two trials [30, 35] because they did not report all the outcomes that they said they measured. The checklist for the risk assessment has been attached (S2 Appendix in S1 File).

Findings from the risk of bias assessment revealed that all three out of the four studies [24, 26, 27] classified as quasi-experimental studies had a low risk of bias and one study [25] had an unclear risk of bias. This was because it was unclear whether there were multiple measurements of the outcome for both the pre-and post-test intervention, plus the researchers also failed to specify whether the follow-up was complete and if there were any differences between groups in terms of their follow-up.

All four studies were unclear as to whether the outcomes of the participants included in the comparisons were measured in a similar manner. Nonetheless, it is important to note that the outcomes seemed reliable judging by the number of raters and the intra-rater and inter-raters reliabilities where reported [24–27]. All studies applied appropriate statistical analyses.

## Data synthesis

The research questions were used to structure the analysis of the main themes of the data synthesis. These included (1) effectiveness of PA intervention, (2) key components of the PA intervention, and (3) the relationship between the characteristics of programme that were related to effectiveness on PA. These themes are discussed further below:

**Effectiveness of the PA intervention.** Effectiveness of the PA intervention, measured by the percentage of students who increased their physical activity, was reported in all the 17 studies. In fifteen of the studies reviewed [24–27, 29–38, 40], the assessment was based on surveys. In two of the studies the assessment used a pedometer [28, 39]. Of the 17 studies, eleven reported that there was an increase in the PA levels of the participants in the intervention groups compared to the control group but in only eleven of these studies was the increase in the PA levels statistically significant [24–26, 31–34, 39]. In these studies, participants were

aged between nine and 18 years. The eleven studies reporting statistically significant findings accounted for a combined total of 12,689 students. For the remaining studies, one study reported that there was no change in attitudes, knowledge, perceptions of PA or in PA level [35]. One study reported different findings between its male and female participants [30]. PA levels and social support differed significantly between males and females. According to the research, during the intervention, boys had higher PA levels compared to girls, but social support was significantly higher in girls. But during the follow-up, i.e., after three months, statistically significant findings were reported for both boys and girls in the intervention group compared to the comparison group [30], implying that levels of PA had increased for both boys and girls in the intervention group during the follow-up. This study [34] reported changes in knowledge, attitude, and behaviour in both the intervention and comparison groups although the increments were not statistically significant and one study [40] reported no significant improvement in BMI but immediate significant changes were seen in PA, sedentary behavior, and body image satisfaction among the control group.

In terms of the magnitude of effect, six studies [28–30, 33, 39] reported that students in intervention groups were three to six times more likely to be physically active both within and outside school compared to students in the control groups.

**Critical components of the PA intervention.**   This review assessed the components of the PA intervention in terms of duration of intervention including follow-up, time (per day or per week) spent implementing the intervention, and implementation type, e.g., theory-based educational lessons on PA or engaging in PA (PE / sports / exercises, etc.).

*Duration.* Only two studies failed to specify the 'duration' of their interventions [34, 35]; in the rest of the studies, duration of implementation ranged from six weeks to three years. Only twelve studies specified their follow-up duration [24, 26, 28, 30, 32–36, 38–40], the rest were unclear as to whether they had any follow-ups; the duration of follow-up in these studies ranged from two months to one year.

The intervention was in the form of, theory-based educational lessons in ten studies [24–26, 30, 31, 33, 35–37, 40]. On the other hand, engaging in PA was used in only two studies [29, 39]. Both theory-based educational lessons and engaging in PA were applied in three studies [28, 32, 34]. At least five studies [24, 26, 31, 35, 37] failed to specify the time spent per session / day implementing the PA interventions. The time spent in the rest of the studies ranged from 15 minutes to 90 minutes per day.

In terms of intervention duration, all the studies that reported statistically significant increments in PA levels had a duration of not less than three months [26, 28, 33] (some up to three years, e.g., [25]) with maintenance of not less than 60% in PA levels in two interventions that continued over three years. [25, 26]. The follow-up period for these studies ranged from six months [28, 33], to one year [26] On the other hand, three studies [30, 34, 35] that reported no change or insignificant findings had only up to three months of intervention and follow-up, implying that a shorter duration of school-based intervention might be ineffective when it comes to increasing PA levels among school students.

*Intensity (time and frequency per day or week).* In terms of the intervention intensity, all studies that reported statistically significant increments in PA levels lasted at least 60 minute per day [28, 31, 32] or 3 hours / week [29]. One study reported insignificant findings and another no change [34] in PA levels with regards to time spent in a day implementing the PA intervention. In [30], a healthy lifestyle course was implemented for 15 minutes a day, five times a week, for six weeks and in [34], the researchers implemented 15 minutes of role-play for an unspecified 'duration' followed by a 15 minutes small group discussion and followed by 30 minutes problem solving exercise and game-playing exercise.

### Intervention content: PA interventions

It should be noted that the 'implementation type' in [30, 34] was theory-based educational lessons on PA. The rest of the studies focusing on this type of implementation reported statistically significant improvements in PA levels. The two-study focusing on engaging in PA alone reported a statistically significant improvement in PA levels following the intervention ($p < 0.005$) El Ansari et al., and Allafi, reported statistically higher step 3,429 (SD 458) in the intervention group comparing the control group and two out of three studies focusing on both educational lessons on PA and engaging in PA [28, 32] reported statistically significant improvement in PA levels following the intervention. These studies reveal the significance of longer 'time' and 'duration' of interventions in increasing PA levels among school-aged children.

In most of the studies, the interventions were delivered by the teachers, but the training that they had was not reported. Other professionals that implemented the programmes included nutritionists and counsellors.

### Intervention content: Combined PA and healthy eating interventions

It should be noted that the only combination used together with PA (educational lessons and exercise) in the studies reviewed was healthy eating, i.e., improvement of dietary behaviour. The effectiveness of this combination on PA was tested in seven studies [24–27, 31, 34, 35, 37]. Two [34, 35] of these seven studies reported insignificant increments in the levels of PA when using a combination of different school-based programmes in the intervention groups ($p > 0.05$). Three [24–26] of the five remaining studies had two intervention groups, i.e., one where the intervention was PA alone (both educational lessons and exercise) and the other where the intervention was both PA and a healthy eating programme. In the [24], both PA only group and the PA and healthy eating group had 965 students each. After the intervention, improvements in PA levels in the PA only group was 56%, whereas improvements in PA levels in the PA and healthy eating group was 67% ($p < 0.005$) [24].

The two remaining studies [31, 37] also reported statistically significant improvements ($p < 0.05$) in PA levels among the participants in the intervention groups; they did not compare a PA intervention and a combined PA and healthy eating combined.

## Discussion

The purpose of this review was to assess the effectiveness of school-based PA interventions for increasing PA among schoolchildren in Middle Eastern and Arabic-speaking countries. We also aimed to identify the critical components required in a school-based intervention programme to increase physical activity among this target population and identify whether particular combinations of elements of school-based programmes are more effective than others in increasing physical activity. Overall, eleven of 17 studies reported statistically significant improvements in the levels of PA among their participants. The studies with a follow-up period greater than three months reported sustained PA levels. The effect size for self-reported increase in physical activity ranged from 58%, [33], to 72% [28]. Although the circumstances in the two trials did not substantially vary, there is a noticeable difference in impact, as shown by the effect sizes.

Furthermore, PA intervention increased participants' knowledge of motor performance by 64% in one study [33]. For the remaining four studies [30, 34, 35, 40], they did not find any statistically significant improvement PA levels. The ineffectiveness of this particular programme during its intervention phase may be attributed to the fact that the duration was only three months. Levels of PA were likely to be closely related to factors such as the content and details

of the school-based intervention programme. Overall, durations of not fewer than three months, and 60 to 90 minutes per day for at least three days a week, were the most effective.

This suggests that the programme duration should be no fewer than three months, and an intensity of 60 to 90 minutes per day for at least three days a week. The studies did not report any statistically significant differences when comparing between schoolteachers and other professions in implementing the programmes, but one study illustrated those schoolteachers with appropriate training and equipped with the right skills, resources and information can have a positive influence on the behaviour of students, and they can deliver the intervention with the highest fidelity [36]. Multi-component programmes, i.e. those that take into consideration PA, health and diet, were shown to ensure that students understand the significance of these three factors [38].

This review suggested PA for at least 60 minutes per day is required for a programme to be effective. In a number of studies assessing the intensity of PA [29, 32, 33], intensity was found to be significantly associated with an increase in moderate-to-vigorous PA. One study [33] reported that learners engaged in this level of activity for around 30–50% of physical education (PE) class time, but greater percentages of moderate-to-vigorous PA were recorded when PA occurred during break and lunchtimes and in other lessons, as well as during extracurricular activities. This implies that even though PE can be effective in promoting PA, one lesson alone might not be fully effective and so schools need to consider the potential for PA at other times.

This review found that most of the programmes that have been evaluated were multi-component (lifestyle, diet, education) and few were focused solely on PA intervention, i.e., exercise and education about PA/exercise. However, there are a number of global school-based PA interventions developed and implemented in different countries to promote PA in schools that have not been implemented to any extent in Arabic speaking countries such as The Daily Mile program, Let's Move and Active Schools which they have been very effective in increasing the level of PA, physical fitness and reducing inactive time among schoolchildren [41–43].

Likewise, Barbosa Filho et al. [44] carried out an umbrella systematic review focusing on low and middle-income countries. They have reviewed fifty reviews and twenty-five studies, to synthesis evidence for the effectiveness of PA programs interventions among children and adolescents aged 6 to 18 years old. This umbrella review confirms that PA interventions can promote PA behavior among this population, also the researchers concluded that there was strong evidence school-based PA interventions can impact positively on the PA level among young people. Moreover, their findings are in line with other reviews conducted internationally [45–48]. However, it should be noted that due to unique climatic, social, and cultural factors in Arabic-speaking countries, it is unlikely that interventions developed elsewhere will be applied directly or without modification to those countries [49].

The findings of this review shed light on several aspects to consider for PA programs in the future to recommend enhanced robustness of PA interventions. First, the number of published studies in Arabic-speaking countries that met the inclusion criteria (n = 17), which constitutes only 30% of Arabic-speaking countries (n = 7 countries). The limited number of studies focusing on this population points to the fact that research into this topic in the region is still in its infancy. These results align with those of Nash et al. whose focused only on six GCC countries and Benajiba et al., which recently focused on interventions to promote physical activity in the region [19, 50].

Previous systematic reviews and meta-analyses synthesised international evidence for the effectiveness of school-based PA programmes among adolescents [51–54]. One previous review concluded that moderate-to-vigorous PA can be improved by at least 60 minutes of physical education per day when implemented in school curriculums [54]. In these studies, multi-component interventions also had the largest significant effect on PA levels, consistent

with the concept of comprehensive school PA programmes (CSPAPs) recommended by the World Health Organization (WHO) and the Centres for Disease Control and Prevention (CDC) [53]. Such programmes aim to intervene before children are in school, and when in school through PE and staff involvement.

## Limitations of the evidence included in the review

More school-based PA intervention studies in the region are needed since we found studies in only seven countries amongst 20 countries in the region. Furthermore, most measured only self-reported PA which is vulnerable to significant reporting biases. The short-term nature of most of the identified studies means there is generally a lack of evidence for the sustainability of PA interventions or the longevity of outcomes. Nonetheless, some of the studies had a long duration and/or follow up, e.g. up to three years, and the researchers were able to confirm the sustainability of the PA interventions. More research is needed to explore the relationship between sustainability and impacts, or demographic factors such as socioeconomic position and age.

## Strengths and limitations of the review

The key strength of this review is that it is the first to systematically assess the effectiveness of school-based physical activity interventions for increasing PA among schoolchildren aged 6 to 18 years old in Middle Eastern and Arabic-speaking countries. We included both Arabic speaking countries and additional countries in the same region (Iran, Israel, Turkey, and Cyprus) to maximise the number of potentially relevant studies included in the review and increase to generalizability of findings to this wider area. This resulted in the inclusion of five relevant studies conducted in Iran. Other reviews have previously investigated the effectiveness of school-based PA interventions among children and adolescents in Arabic speaking Countries [55] and in six Gulf countries [50] but this review offered a much more comprehensive narrative synthesis of included articles, adhering to PRISMA guidelines for conducting systematic reviews (S1 Checklist in supplementary material). However, it also has some limitations. First, publications other than journal articles, ie "grey literature". such as books, theses, etc., were not included so it is conceivable that additional school-based interventions with unpublished evaluations were excluded. Furthermore, even though numerous other outcomes have been evaluated in regard to physical activity programmes, this review focused only on the effectiveness of interventions in increasing PA, rather than on other potential associated benefits from the activity.

## Implications for research and practice

Our review demonstrates that some programs have the potential to increase PA levels in school-aged children. The studies with a follow-up period greater than three months reported sustained PA levels. However, long-term school-based interventions combined with rigorous theoretical and methodological frameworks are necessary to determine the true impact of these interventions on children and adolescent in Arab speaking countries.

The findings of this systematic review confirm that school-based interventions that attempt to encourage physical activity among children and adolescents are effective, at least in the short term. These results may provide further support for school-based efforts that aim to enhance students' physical activity levels, bearing in mind the favourable benefits of PA on health in general. Future studies should be more specific in their reporting of descriptions of control group activities and if they are supplemented or replaced by regular physical education sessions. Our results further indicate the need for increased high-quality physical activity

programmes in Arabic-speaking countries in the school environment.; standardised reporting of programme implementation and evaluation would add to the evidence base to support guidance for schools on how best to promote PA. It is necessary to better understand both the mechanisms of action and the assessment of the implementation process to have a deeper understanding of how these types of interventions operate and how they may be implemented in practice.

## Conclusions

This review adds to the existing body of research about the efficacy of school-based interventions to enhance the number of students in school who increase participation in physical activity levels. To date, a limited range of types few evaluations focused solely on assessing PA specific interventions and most of the interventions were multi-component (including education components on lifestyle, and diet, education). Long-term school-based interventions combined with rigorous theoretical and methodological frameworks are necessary to develop, implement and evaluate PA interventions for children and adolescents in Arabic speaking countries. Also, future work in this area must also consider the complex systems and agents by which physical activity is influenced and whether programmes that have been effective in other regions could be adapted for the Arabic speaking region.

## Supporting information

**S1 Checklist. PRISMA 2020 checklist.**
(DOCX)

**S1 File.**
(DOCX)

## Author Contributions

**Supervision:** Lindsay Blank, Elizabeth Goyder.

**Writing – original draft:** Abdullah Alalawi.

**Writing – review & editing:** Elizabeth Goyder.

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
