## [Decision Letter · Decision Letter 0]

23 Mar 2023

PONE-D-23-05274School-based Physical Activity Interventions among Children and Adolescents in the Middle East and Arabic Speaking Countries: A Systematic ReviewPLOS ONE

Dear Dr. Alalawi,

Thank you for submitting your manuscript to PLOS ONE. After careful consideration, we feel that it has merit but does not fully meet PLOS ONE’s publication criteria as it currently stands. Therefore, we invite you to submit a revised version of the manuscript that addresses the points raised during the review process.

We look forward to receiving your revised manuscript.

Kind regards,

Bojan Masanovic, Ph.D.

Academic Editor

PLOS ONE

Journal Requirements:

Reviewers' comments:

Reviewer's Responses to Questions

**Comments to the Author**

1. Is the manuscript technically sound, and do the data support the conclusions?

Reviewer #1: Yes

Reviewer #2: Partly

2. Has the statistical analysis been performed appropriately and rigorously? 

Reviewer #1: Yes

Reviewer #2: N/A

3. Have the authors made all data underlying the findings in their manuscript fully available?

Reviewer #1: Yes

Reviewer #2: Yes

4. Is the manuscript presented in an intelligible fashion and written in standard English?

Reviewer #1: Yes

Reviewer #2: No

5. Review Comments to the Author

Reviewer #1: Dear authors, the research topic is very current. The manuscript is methodologically very well written, in accordance with the guidelines for writing review research. The manuscript has some details that need fixing:

1. The first sentence in the Introduction requires a citation

2. in the section Selection of studies, it is unnecessary to state the author's initials, this part should be written without a specific indication of the author

3. In Figure 1, the number 1349 needs to be technically corrected

4. #the combined number of school-aged children who took part in the 17 studies was 15,551 with a mean sample size of 1,113" - I am not clear about this average? That is not correct information! And it is not necessary because each study has its own number of subjects.

Reviewer #2: Manuscript number: PONE-D-23-05274

Thank you for the opportunity to review the paper entitled: ‘School-based physical activity interventions among children and adolescents in the Middle East and Arabic speaking countries: a systematic review.

The manuscript addresses an important gap in the literature in relation to children’s physical activity in Arabic and Middle Eastern countries. The paper is reasonably well written but there are several typographical and grammatical errors throughout. It is not clear if all the PRISMA guidelines were adhered to, please see my comments in the methods section of this review. I have listed several suggestions below. I hope the authors find the comments helpful and wish them all the best with their future research.

Abstract

i) Page 3 – Conclusion- check grammar in section “multi-component includes education”

Introductory section

i) Please add references to all sentences which do not have a reference in the introduction section of the manuscript.

ii) I wonder, given that this article solely focuses on children, it might be better to acknowledge in one sentence the health impact of low levels of PA on the broader population (e.g., adults) but then focus more specifically on the health impact of low levels of PA in children, as there is quite a lot of literature in this area now.

iii) Start of second paragraph suggest using ‘increase’ rather than ‘raise’

iv) Some of the references used are quite old and should be updated – suggest reviewing all references that are older than 10 years.

v) Suggest indicating what the PA levels are in Arabic countries to provide more insight in the sentence: More specifically, the average level of PA is low in the Arabic speaking countries, compared with other countries [7].

vi) Last sentence of paragraph 2 - Suggest that exercise and physical activity are not used interchangeably, as people can still be physically active without exercising. Suggest checking the manuscript throughout and changing where necessary.

vii) Suggest that physical inactivity is not used interchangeably with sedentary behaviour. As being physically inactive means that someone is not doing enough physical activity and being sedentary means that they are sitting or lying down for long periods.

viii) Last sentence on page 5 - suggest changing the word ‘find’ to: understand

Methods

i) Did you use the PRISMA framework for systematic reviews? If you used the PRISMA guidelines, suggest that this be added to the methods and abstract.

ii) Page 7 - Add ‘the’ to the sentence commencing: Search parameters were limited to ‘the’ English language.

iii) In the section entitled: Search Strategy – I suggest combining individual sentences into one paragraph. I am just wondering why the Arabic studies data bases were not included in the search.

Selection of studies

iv) The study selection does not seem to have followed the PRISMA guidelines where 2 to 3 reviewers independently select relevant studies.

v) Was a referencing packaged used (e.g., Endnote)? Please describe how the references were managed.

vi) MENA should be written in full when first cited in the paper with MENA in brackets.

Study characteristics

vii) Page 10 second paragraph second sentence. Note: 8-year-olds are not generally considered adolescents – those 13 years and above are adolescents. Suggest revising this section

Risk of Bias

viii) Page 16 first sentence suggest adding ‘an’ to the following part of the sentence had ‘an’ unclear risk of bias.

ix) Page 16 first paragraph second last sentence: Suggest rewording this sentence to state that selective bias was low in ten trials and unclear in two trials …..

x) In relation to the above sentence – it seems that one trial is missing??

xi) Page 16 second paragraph. This paragraph includes one very long sentence – suggest revising to at least two sentences.

xii) Page 16 - suggest providing more details of the relevant studies in relation to the following sentence: Nonetheless, it is important to note that the outcomes seemed reliable judging by the number of raters and the intra-rater and inter-raters reliabilities where reported.

Data synthesis

xiii) Page 17 second sentence check grammar for ‘These includes’

xiv) Page 17 – need a full stop in this sentence: PA levels and social support differed significantly between males and females According to the research, during the intervention…

xv) Also suggest removing the word ‘an’ from this sentence: This study [27] reported an changes in knowledge..

xvi) Page 18 check grammar in this sentence: Engaging in PA, on the other hand, was used in only two study

And

Page 18 The two study focusing

xvii) I am not 100% sure of the journal formatting but suggest that the authors names should be referred to in sections on page 19 where it just uses numbers e.g., alone (22) reported….

xviii) Page 20 add ‘the’ to this sentence: In [17], both PA only

Discussion

i) Page 21 second paragraph of the discussion. It is unclear which studies are being referred to in this paragraph. Were the four remaining studies a part of the first study? Suggest revising and adding references to this paragraph.

ii) Also suggest adding more detail in relation to the following sentence from the above paragraph: related to factors such as the content and details of the school-based intervention programme.

iii) Page 22 suggest adding ‘the’ to the sentence: to consider potential for PA other times.

iv) Page 22 third paragraph suggest adding ‘that’ to the following sentence: schools have not been implemented to any extent

v) Page 23 first paragraph suggest adding ‘that’ to this sentence: strong evidence school-based PA interventions

vi) Page 23 paragraph 1 – Suggest revising the last sentence for grammar.

vii) Page 23 paragraph 2 -suggest revising this part of the sentence: ‘met inclusion criteria’ to read: that met the inclusion criteria

viii) Page 23 paragraph 2 Suggest expanding a little on the following sentence so that the reader understands the focus of these reviews in comparison to the current review in the section: These results align with those of Nash et al. and Benajiba et al., which recently focused on interventions to promote physical activity in the region [43,44].

Implications for research

i) Page 25 first paragraph: suggest revising this sentence: ‘have potential in increasing PA levels’ to read have the potential to increase PA levels

ii) Page 25 paragraph two suggest keeping the PA focus and stating: favourable benefits of PA on health in general

and

iii) Suggest removing capitalization on the word: Standardised.

Conclusion

Suggest revising the conclusion for grammatical errors.

i) Page 26: conclusion – check grammar in this section: ‘who increase participate in physical activity levels’.

And this section:

ii) To date, limited range of types few evaluations focused solely on assess PA specific interventions and most of the interventions were multi-component (includes education components on lifestyle, and diet, education).

And

iii) An ‘r’ is missing from this section: ‘have been effective in other egions’

6. PLOS authors have the option to publish the peer review history of their article (what does this mean?). If published, this will include your full peer review and any attached files.

Reviewer #1: No

Reviewer #2: **Yes: **Dr Anne-Maree Parrish

---

## [Author Response · Author response to Decision Letter 0]

10 May 2023

Dear reviewers, many thanks for your valuable feedback, please find in the attachment three documents (Response to Reviewers, Revised Manuscript with Track Changes, and Revised Manuscript Clear Version).

---

## [Decision Letter · Decision Letter 1]

23 May 2023

PONE-D-23-05274R1School-based Physical Activity Interventions among Children and Adolescents in the Middle East and Arabic Speaking Countries: A Systematic ReviewPLOS ONE

Dear Dr. Alalawi,

Thank you for submitting your manuscript to PLOS ONE. After careful consideration, we feel that it has merit but does not fully meet PLOS ONE’s publication criteria as it currently stands. Therefore, we invite you to submit a revised version of the manuscript that addresses the points raised during the review process.

We look forward to receiving your revised manuscript.

Kind regards,

Bojan Masanovic, Ph.D.

Academic Editor

PLOS ONE

Reviewers' comments:

Reviewer's Responses to Questions

**Comments to the Author**

1. If the authors have adequately addressed your comments raised in a previous round of review and you feel that this manuscript is now acceptable for publication, you may indicate that here to bypass the “Comments to the Author” section, enter your conflict of interest statement in the “Confidential to Editor” section, and submit your "Accept" recommendation.

Reviewer #1: All comments have been addressed

Reviewer #3: (No Response)

2. Is the manuscript technically sound, and do the data support the conclusions?

Reviewer #1: Yes

Reviewer #3: Partly

3. Has the statistical analysis been performed appropriately and rigorously? 

Reviewer #1: Yes

Reviewer #3: N/A

4. Have the authors made all data underlying the findings in their manuscript fully available?

Reviewer #1: Yes

Reviewer #3: No

5. Is the manuscript presented in an intelligible fashion and written in standard English?

Reviewer #1: Yes

Reviewer #3: Yes

6. Review Comments to the Author

Reviewer #1: All comments from the reviewers have been accepted and the manuscript has now been reconciled with the comments submitted.

Reviewer #3: This was an interesting review that explores physical activity interventions across Arabic-speaking countries. The manuscript is certainly unique in its own right, however, there are some major concerns worth raising.

My primary concern with this review involves the Methods section.

Firstly, the use of additional electronic academic databases would likely have added to the study in both complexity and sample size. If the scope of this study were expanded to use additional databases, more sources might have been identified and explored. For example, the following electronic databases could have been used for a more thorough and inclusive search;

“ArticleFirst; Biomed Central; BioOne; BIOSIS; EBSCOHost; JSTOR; ProQuest; SAGE Reference Online; ScienceDirect; SpringerLink; Taylor & Francis; and Wiley Online.” These databases would have likely added to the overall literature search in their academic rigor, aim, and biomedical scope. While PubMed/MEDLINE, Web of Science, Scopus and CINAHL is an excellent database, the use of more databases would have added to the study sample size. Also, I recommend adding French in the language search as well. Not using French would potentially neglect countries like Morocco, Tunisia, Algeria etc. as part of the search parameters.

Secondly, the Arab world (or the Middle east) should be clearly and expressly defined. I strongly suggest the authors review this article thoroughly and clearly define what ‘Arabic-speaking countries’ are. What defines the ‘Middle East’? Israel (Arabic is a recognized language in Israel), Iran, and Turkey may be argued as part of the Middle East…were those counted as part of the search parameters. I recommend using the Arab League member states as the definition. This should be written clearly in the methods section. I suggest citing the article too to improve the merit of this review.

https://www.ncbi.nlm.nih.gov/pmc/articles/PMC3988371/

7. PLOS authors have the option to publish the peer review history of their article (what does this mean?). If published, this will include your full peer review and any attached files.

Reviewer #1: **Yes: **Jovan Gardasevic, University of Montenegro

Reviewer #3: **Yes: **Basil H. Aboul-Enein

---

## [Author Response · Author response to Decision Letter 1]

30 May 2023

A response to the reviewer’s comments point by point and how we addressed them: 

Reviewer #1 Comments: 

Reviewer #1: All comments from the reviewers have been accepted and the manuscript has now been reconciled with the comments submitted.

Thanks. 

Reviewer #3: This was an interesting review that explores physical activity interventions across Arabic-speaking countries. The manuscript is certainly unique in its own right, however, there are some major concerns worth raising.

My primary concern with this review involves the Methods section.

Firstly, the use of additional electronic academic databases would likely have added to the study in both complexity and sample size. If the scope of this study were expanded to use additional databases, more sources might have been identified and explored. For example, the following electronic databases could have been used for a more thorough and inclusive search;

“ArticleFirst; Biomed Central; BioOne; BIOSIS; EBSCOHost; JSTOR; ProQuest; SAGE Reference Online; ScienceDirect; SpringerLink; Taylor & Francis; and Wiley Online.” These databases would have likely added to the overall literature search in their academic rigor, aim, and biomedical scope. While PubMed/MEDLINE, Web of Science, Scopus and CINAHL is an excellent database, the use of more databases would have added to the study sample size. Also, I recommend adding French in the language search as well. Not using French would potentially neglect countries like Morocco, Tunisia, Algeria etc. as part of the search parameters.

We appreciate the concern that the limitations of our database search strategy and restriction to English language articles may have led to our searches not being exhaustive but were constrained by resource considerations as this review was undertaken by the first author within a PhD programme. We have mitigated these limitations in several ways including forward and backwards (citation and reference) searching and checking the reference lists of other related systematic reviews. These strategies provide reassurance that it is unlikely that there is a substantive number of additional studies which would meet our inclusion criteria and change our overall conclusions about the nature of current evidence in this field. We have added a more detailed explanation in the Discussion under “Limitations” on the specific limitations and associated mitigating strategies. 

“Despite the limitations of our initial database searches which were constrained by available resources, we have mitigated these limitations in several ways including forward and backwards (citation and reference) searching and checking the reference lists of other related systematic reviews. These strategies provide reassurance that it is unlikely that there is a substantive number of additional studies which would meet our inclusion criteria and change our overall conclusions about the nature of current evidence in this field.”

Secondly, the Arab world (or the Middle east) should be clearly and expressly defined. I strongly suggest the authors review this article thoroughly and clearly define what ‘Arabic-speaking countries’ are. What defines the ‘Middle East’? Israel (Arabic is a recognized language in Israel), Iran, and Turkey may be argued as part of the Middle East…were those counted as part of the search parameters. I recommend using the Arab League member states as the definition. This should be written clearly in the methods section. I suggest citing the article too to improve the merit of this review.

https://www.ncbi.nlm.nih.gov/pmc/articles/PMC3988371/

Thank you for this suggestion; we recognize the need for clarification of both the inclusion criteria and the rationale for the choice of countries included as this is important for the interpretation of our findings and their generalizability to other countries in the region.

We have therefore added additional clarification on pages 5 and 7 of the manuscript

Methods:

“We included Arabic-speaking countries located in the Arabian Peninsula, northern parts of Africa, and the East Mediterranean [13]. It contains 22 countries ((i.e., Yemen, United Arab Emirates, Syria, Saudi Arabia, Qatar, Oman, Lebanon, Kuwait, Jordan, Bahrain, Palestinian, Tunisia, Egypt, Libya, Morocco, Comoros, Djibouti, Mauritania, Algeria, Sudan, Somalia, and Iraq). These countries are defined as the 22 member countries of the League of Arab States [14]; along with countries in the Middle East that are not counted in the Arabic-speaking countries (i.e., Iran, Israel, Turkey, and Cyprus).”

Discussion (under Strengths):

“We included both Arabic speaking countries and additional countries in the same region (Iran, Israel, Turkey, and Cyprus) to maximise the number of potentially relevant studies included in the review and increase to generalizability of findings to this wider area. This resulted in the inclusion of five relevant studies conducted in Iran.”

Thank you so much for your valuable suggestions.

---

## [Decision Letter · Decision Letter 2]

20 Jun 2023

School-based Physical Activity Interventions among Children and Adolescents in the Middle East and Arabic Speaking Countries: A Systematic Review

PONE-D-23-05274R2

Dear Dr. Alalawi,

We’re pleased to inform you that your manuscript has been judged scientifically suitable for publication and will be formally accepted for publication once it meets all outstanding technical requirements.

Kind regards,

Bojan Masanovic, Ph.D.

Academic Editor

PLOS ONE

Additional Editor Comments (optional):

Reviewers' comments:

Reviewer's Responses to Questions

**Comments to the Author**

1. If the authors have adequately addressed your comments raised in a previous round of review and you feel that this manuscript is now acceptable for publication, you may indicate that here to bypass the “Comments to the Author” section, enter your conflict of interest statement in the “Confidential to Editor” section, and submit your "Accept" recommendation.

Reviewer #1: All comments have been addressed

Reviewer #3: All comments have been addressed

2. Is the manuscript technically sound, and do the data support the conclusions?

Reviewer #1: Yes

Reviewer #3: Yes

3. Has the statistical analysis been performed appropriately and rigorously? 

Reviewer #1: Yes

Reviewer #3: N/A

4. Have the authors made all data underlying the findings in their manuscript fully available?

Reviewer #1: Yes

Reviewer #3: Yes

5. Is the manuscript presented in an intelligible fashion and written in standard English?

Reviewer #1: Yes

Reviewer #3: Yes

6. Review Comments to the Author

Reviewer #1: All comments have been acknowledged by the authors, the manuscript has been revised and is now ready for publication.

Reviewer #3: thank you very much for addressing the recommendations and suggestions. the manuscript has been improved.

7. PLOS authors have the option to publish the peer review history of their article (what does this mean?). If published, this will include your full peer review and any attached files.

Reviewer #1: **Yes: **Jovan Gardasevic

Reviewer #3: **Yes: **Basil H. Aboul-Enein

---

## [Editor Report · Acceptance letter]

22 Jun 2023

PONE-D-23-05274R2 

School-based Physical Activity Interventions among Children and Adolescents in the Middle East and Arabic Speaking Countries: A Systematic Review 

Dear Dr. Alalawi:

I'm pleased to inform you that your manuscript has been deemed suitable for publication in PLOS ONE. Congratulations! Your manuscript is now with our production department. 

Kind regards, 

on behalf of

Dr. Bojan Masanovic 

Academic Editor

PLOS ONE